# The Risk of Bleeding and Adverse Events with Clopidogrel in Elective Hip and Knee Arthroplasty Patients

**DOI:** 10.3390/jcm11071754

**Published:** 2022-03-22

**Authors:** Cheng-Ta Wu, Tzu-Hsien Lien, I-Ling Chen, Jun-Wen Wang, Jih-Yang Ko, Mel S. Lee

**Affiliations:** 1Department of Orthopaedic Surgery, Kaohsiung Chang Gung Memorial Hospital, Kaohsiung 833, Taiwan; oliverwu429@cgmh.org.tw (C.-T.W.); wangjw@cgmh.org.tw (J.-W.W.); kojy@cgmh.org.tw (J.-Y.K.); 2Department of Family Medicine, E-Da Hospital, Kaohsiung 824, Taiwan; sunshawn1986@yahoo.com.tw; 3Department of Pharmacy, Kaohsiung Chang Gung Memorial Hospital, Kaohsiung 833, Taiwan; iling10@cgmh.org.tw

**Keywords:** antiplatelet, clopidogrel, total knee arthroplasty, total hip arthroplasty, thromboembolism, cardiocerebral events

## Abstract

Orthopedic surgeons often face a clinical dilemma on how to manage antiplatelet therapies during the time of surgery. This retrospective study is aimed to investigate the bleeding risk and adverse events in patients who hold or keep clopidogrel during elective major joints arthroplasty. Two hundred and ninety-six patients that were treated with clopidogrel while undergoing total hip or knee joint replacement between January 2009 and December 2018 were studied. Group 1 included 56 patients (18.9%) who kept using clopidogrel preoperatively. Group 2 included 240 patients who hold clopidogrel use ≥5 days preoperatively. Blood transfusion rates, estimated blood loss, complication rates, and adverse cardiocerebral events were collected and analyzed. The mean total blood loss was more in the group 1 patients as compared with that in the group 2 patients (1212.3 mL (685.8 to 2811.8) vs. 1068.9 mL (495.6 to 3294.3), *p* = 0.03). However, there was no significant difference between the two groups of patients regarding transfusion rates, bleeding-related complications, and infection rates. There was a trend toward a higher incidence of adverse cardiocerebral events in patients withholding clopidogrel for more than 5 days before surgery. The results of this study suggest that clopidogrel continuation could be safe and advisable for patients at thrombotic risk undergoing primary major joint replacement. Acute antiplatelet withdrawal for an extended period of time might be associated with an increased risk of postoperative thromboembolic events. More studies are required in the future to further prove this suggestion.

## 1. Introduction

Total hip (THA) and total knee (TKA) arthroplasties are widely performed to treat painful end-stage osteoarthritis. A considerable proportion of these elderly THA and TKA patients takes antiplatelet agents for primary or secondary prevention of either cardiovascular or cerebrovascular events [1]. According to the literature, perioperative cardiovascular complications are an important source of morbidity and mortality for more than 200 million patients worldwide who underwent non-cardiac surgery each year [2,3]. In patients with a history of acute myocardial infarction (AMI), the incidence of post-TKA myocardial infarction was about 1.7% [4,5]. Therefore, orthopedic surgeons face a clinical dilemma on how to manage these antiplatelet therapies during the time of surgery. Excessive bleeding in hip and knee arthroplasties may increase the risk of wound complications, infection, and more transfusion [6,7]. Instead, acute withdrawal of the therapies for an extended period of time poses the life-threatening danger of vascular thrombosis on patients [8,9]. Rapid return of platelet aggregation and the platelet rebound phenomenon were found in the setting of acute antiplatelet withdrawal [10,11]. Clinical discretion is needed to weigh the risk of bleeding if the antiplatelet therapy is continued perioperatively against the risk of a cardiovascular or cerebrovascular accident if it is discontinued.

The American Academy of Orthopedic Surgeons in its 2011 guidelines recommended discontinuing anti-platelet therapy before undergoing elective total joint arthroplasty [12]. However, the literature on which the recommendation was based was from studies on coronary artery bypass surgeries instead of orthopedic surgeries [13,14]. In 2012 the 9th American College of Chest Physicians (ACCP) guidelines on perioperative management of antithrombotic therapy also proposed stopping antiplatelet agents 7~10 days before surgery [15]. They suggested, however, the continuation of aspirin around the time of surgery in patients with moderate to high risk of cardiovascular events. Because only a few studies have assessed the effect of perioperative continuation of clopidogrel, the ACCP guidelines have not given recommendations on its use. Other studies on clopidogrel gave conflicting information on the perioperative bleeding risk and complications, with most of them suggesting discontinuing its use 5~10 days preoperatively [6,16,17].

Studies from UK and Scotland have demonstrated that there were a wide variety of practices among orthopedic surgeons regarding the continuation or cessation of clopidogrel perioperatively [18,19]. Since the existing literature landed little insight on perioperative management of clopidogrel during hip and knee arthroplasties, the main objective of this retrospective study is to analyze the bleeding risk, surgical complications, and major adverse cardiocerebral events (MACCE) between patients who hold or keep clopidogrel in elective THAs and TKAs.

## 2. Methods

### 2.1. Study Design

A total of 9521 primary TKA and THA patients operated on at our orthopedic department between January 2009 and December 2018 were searched in the electronic medical system. Based on the inpatient/outpatient pharmacy database, all the patients on clopidogrel undergoing TKA or THA were identified. The inclusion criteria comprised patients with ongoing clopidogrel during the perioperative period, a detailed evaluation by their prescribing physicians before elective surgery, a follow-up time for at least 90 days, and complete chart records. Patients with one of the following conditions were excluded, (1) within the critical timeframe of dual antiplatelet therapy post percutaneous coronary intervention (PCI), e.g., ≤1 month after PCI with a bare-metal stent or ≤6 months with a drug-eluting stent, (2) recent stroke in the last 6 months, (3) high risk for thrombotic events and requiring bridging therapy perioperatively, (4) concomitant use of more than one antiplatelet therapy or other anticoagulants such as warfarin, (5) revision arthroplasties or unicompartmental arthroplasty, (6) previous joint trauma surgeries that required larger dissection for implant removal or bone graft for reconstruction, (7) a diagnosis other than osteoarthritis and osteonecrosis, (8) abnormal coagulation profiles, or (9) missing data of the last dose of clopidogrel. The study was conducted with a waiver of patient consent but approved by the Institutional Review Board of our institute.

The decision of continuing or the days of withholding clopidogrel before surgery was made based on the individual surgeon’s discretion and preference. The patients were categorized into two groups. Group 1 included patients who kept using clopidogrel preoperatively or did not hold clopidogrel within 5 days before the surgery, and group 2 included patients who hold clopidogrel more than 5 days preoperatively. The demographics of the patients, including the American Society of Anesthesiologists (ASA) score, type of anesthesia, coagulation profiles, and medications used for VTE prophylaxis were recorded. The thromboembolic and the bleeding risks of the patients were evaluated by the CHADS2-VASc score and HAS-BLED score. The tourniquet was applied to all TKAs. Peri-articular tranexamic acid injection as part of the Cocktail formula was routinely used. Postoperative hemoglobin (Hb) levels were recorded on the first, second, and third or fourth days postoperatively. All the surgeons in our department routinely started clopidogrel on postoperative day 1 if there was no complication on the wound. Total blood loss was calculated according to the method proposed by Good et al. and Nadler et al., which used the maximum postoperative reduction in Hb level adjusted by weight and height of the patient and calculated by the following formula: Total blood loss = (Total blood volume × [change in Hb level/preoperative Hb level]) × 1000 + volume transfused [20,21].

The trigger for blood transfusion was also decided by the individual surgeons and was generally set at an Hb level of ≤7 g/dL in healthy patients without anemic signs, ≤9 g/dL in patients with cardiovascular or cerebrovascular diseases, according to Société Française d’Anesthésie Et De Réanimation (SFAR) guidelines, or patients with signs of hypovolemia such as tachycardia or decreased urine output [22].

### 2.2. Outcomes

The primary outcome of the study was to compare the major bleeding risk between the two groups of patients. The bleeding risk was evaluated by any clinically fatal bleeding, critical organ bleeding, the estimated blood loss, and the incidence of blood transfusion.

The secondary outcome of the study was to compare the complication rates between the two groups. These included the wound complications, hematoma that required returning to theater, periprosthetic joint infection (PJI), VTE within 90 days after the surgery, and MACCE. The presence of PJI was based on the Musculoskeletal Infection Society (MSIS) criteria [23]. The MACCE was defined as AMI, heart failure, acute ischemic or hemorrhagic stroke, and transient ischemic attack. Rates of returning to theater for washout, 30-day emergency room (ER) returning after discharge, and 90-day mortality were also analyzed.

### 2.3. Statistical Analysis

The Student’s *t*-test or the Mann–Whitney U test was used for comparison of continuous variables between the two groups in the distribution of demographic data (including age, BMI, and preoperative laboratory data) and total blood loss. The χ^2^ test or Fisher’s exact test was used when analyzing the differences of dichotomous variables between the two groups (including gender, ASA score ≥ 3, types of anesthesia, and numbers of patients with blood transfusion). The results were expressed as the mean ± standard deviation.

The Mann–Whitney U test was used to detect the differences in the Hb levels at different postoperative days, followed by Bonferroni corrections for *p*-values. All tests were two-sided, and *p* < 0.05 was considered significant. All statistical comparisons were made using the Statistical Package for Social Sciences (SPSS) (version 22; SPSS Inc., Chicago, IL, USA).

## 3. Results

As a result, 296 patients (3.1%) who fulfilled the criteria mentioned above were enrolled in the analysis. According to the guidelines and the consensus documents for a proposed scheme of stratification of thrombotic risk, our study population belongs to “low to moderate risk” groups [24,25]. The most common indication of using clopidogrel in the study population was coronary artery disease (55%), followed by cerebrovascular accident (21%) (Figure 1). There were 56 patients (18.9%, 15 THA, and 41 TKA) categorized in group 1. Among them, 30 patients continued clopidogrel throughout the perioperative period, and 26 patients held clopidogrel within 5 days before the surgery. The other 240 patients (81.1%, 27 THA, and 213 TKA) in group 2 ceased clopidogrel more than 5 days before the surgery. The median time frame (25th, 75th quartiles) for discontinuation of clopidogrel prior to surgery was 7 (5, 7) days. There were no significant differences between the two groups of patients in terms of age, ASA status, coagulation profile, CHADS2-VASc score, HAS-BLED score, and most of the comorbidities (Table 1). The type of anesthesia was decided by the anesthesiologists, and there were significantly fewer patients in group 1 receiving spinal anesthesia. The proportion of male patients, end-stage renal disease, arrhythmia/Af, and THA/TKA ratio were higher in group 1.

As for the major bleeding risk, no patients sustained clinically fatal or critical organ bleeding. There were no significant differences in preoperative Hb and postoperative changes of Hb between the two groups of patients (Table 2). The mean estimated blood loss was about 150 mL more for patients in group 1 than that for patients in group 2 (1212.3 mL (685.8–2811.8) vs. 1068.9 mL (495.6–3294.3), *p* = 0.03). The transfusion rate was similar in both groups (19.6% vs. 16.7%, *p* = 0.28). No difference was detected regarding the mean units of blood transfusion. There was no platelet transfusion in the study population.

The perioperative complications were also compared (Table 3). Postoperative wound complications included prolonged discharge, blister formation, hemarthrosis, and skin necrosis. The incidence of these wound complications in group 1 (21.4%) was not significantly higher than those in group 2 (15%) (OR = 1.55, 95% CI = 0.74~3.21, *p* = 0.23). There was no significant difference between the two groups in the risk of PJI, either. As for the incidence of MACCE, there were 13 patients (5.4%) in group 2 complicated with AMI or cerebrovascular accident postoperatively, and only one patient (1.8%) in group 1 had AMI (OR = 0.32, 95% CI = 0.04~2.48, *p* = 0.48). No significant differences were noticed regarding the incidences of returning to theater for washout and 30-day ER returning. There was no patient that developed deep vein thrombosis, but one patient in the group 2 had a high suspicion for pulmonary embolism and was resolved after treatment. One patient with multiple underlying comorbidities in group 2 had mortality at 12 days after surgery due to acute respiratory failure and sepsis related to his underlying comorbidities.

## 4. Discussion

The optimal perioperative management of antiplatelet therapy remained a clinical challenge in total joint arthroplasty. There was no risk assessment tool or clinical rationale available [26]. The continuation or interruption of these medications was decided at the discretion of the surgeon and cardiologist based on the perceived risks of bleeding and thrombosis. Despite the tendency of increasing bleeding risks, aspirin continuation in major joint replacements has been shown to be safe and not necessarily to result in more blood loss [27,28]. Clopidogrel has a different mechanism and pharmacokinetics from aspirin in its antiplatelet effect. Current literature on perioperative management of clopidogrel in joint arthroplasty is limited and is mostly against its continuation because of increased bleeding-related events and transfusion rates [6,16]. In our study, we found the mean estimated blood loss increased by about 150 mL in patients continuing clopidogrel before surgery compared with those who withhold it. However, there was no significant difference between the two groups of patients regarding the length of hospital stay, transfusion rates, wound complications, and infection rates. Albeit not statistically significant, there was a higher incidence of MACCE in patients withholding clopidogrel for more than 5 days before surgery.

Nandi et al. conducted the first study on clopidogrel in major joint arthroplasty. They divided the patients into three groups based on the days of stopping clopidogrel before surgery [6]. The authors reported significantly higher reoperation rates for infection (25%) and more antibiotic use (38%) in patients continuing clopidogrel prior to the surgery. Advanced age, ASA = 4, and revision arthroplasty predicted a higher risk of complications. One of the limitations of this study was relatively small patient numbers in the continuing-drug group (only eight patients), rendering an exceptionally high infection rate. In addition, the varied complexity of surgery secondary to the inclusion of revision and unicompartmental arthroplasty in the study might be a confounder. The other similar study reported that continuing clopidogrel before elective TKA and THA resulted in more blood transfusion during hospitalization (37.5% vs. 15.3%, *p* = 0.02) [16]. There was no difference in the incidence of 30-day adverse cardiac events. This study only used blood transfusion as their primary outcome, and also included revision arthroplasty in the analysis. In comparison, our study included only primary TKA and THA to decrease the confounding of surgical complexity. Moreover, we evaluated bleeding risk by calculating the amount of estimated blood loss in addition to changes of Hb and blood transfusion. While statistical significance was not achieved in our data, continuing clopidogrel at the time of surgery was potentially helpful to the patients against postoperative MACCE without increasing risks of bleeding-related complications.

Most of the findings regarding bleeding risk related to clopidogrel came from the cardiac surgery literature [13,29]. These results and recommendations to withhold therapy have been extrapolated to orthopedic surgeries [1,12,17,30]. Recently, however, a growing body of evidence has shown that clopidogrel could be continued safely without significant increase of hemorrhage or complications throughout the surgery of elderly hip fracture, including bipolar hemiarthroplasty and internal fixation [7,17,19,31]. Earlier hip operations without delay in patients taking clopidogrel mitigated postoperative complications and mortality rate, not necessarily at the expense of more transfusion or blood loss. On the other hand, Collyer et al. found 20.2% of patients, who received clopidogrel therapy and withheld preoperatively, suffered an acute coronary syndrome peaking from 4 to 8 days after withdrawal [32]. Other studies reported that perioperative AMI occurred in nearly 5% of participants within 30 days of noncardiac surgery and preoperative antiplatelet interruption might increase the risk up to 10.2% [33,34]. Tissue trauma, systemic inflammation, and the pro-thrombotic milieu that involved an increment of fibrinogen, plasminogen activator inhibitor, and endogenous catecholamines levels after the surgical stress resulted in a temporary hypercoagulability peaked at postoperative 3~5 days [2,35]. Accordingly, the risk of cardiovascular complications after discontinuation of clopidogrel was highest between postoperative days 4 and 8 [2,26]. It was therefore understandable that continuing perioperative antiplatelet therapy could be helpful to patients at risk of thrombosis in the context of invasive surgeries.

The conventional approach to perioperative management of antiplatelet agents was to discontinue the therapy for 7~10 days [12,30]. Nevertheless, the unpredictable pharmacokinetics of clopidogrel called it into question. Evidence has shown that about 40% of patients are resistant to this drug owing to the genetic polymorphism in the CYP2C19 allele, leading to a large inter-variability in patients’ responsiveness [31,34,36]. Frequent drug-drug interactions such as proton-pump inhibitors, which are often prescribed after major surgeries, also weaken the clopidogrel via competitive inhibition of the CYP2C19 enzyme because both drugs are metabolized by this enzyme [36,37]. In addition, the extrinsic pathway of coagulation cascade initiated by tissue factors after surgical trauma could play an important and compensatory role in surgical hemostasis in the case of a compromised platelet function [1,34]. These are possible explanations on which some studies proposed a low risk of hemorrhage in patients receiving clopidogrel [31]. Despite the fact that it often takes 10 days to replenish the total pool of platelets, it has been shown that normal hemostasis could be achieved if as little as 20% of platelets had normal COX activity [38]. Information from the Food and Drug Administration recommends that platelet aggregation and the bleeding time return to baseline values generally within 5 days after treatment interruption [30]. The traditional drug cessation by 7~10 days hence needs reassessment. A study proposed the discontinuation of therapy before surgery by 48 h, a time frame at which some antiplatelet protection remained but the active metabolites of clopidogrel were cleared so that platelet transfusion could be feasible in case of serious bleeding [37]. According to our results, holding clopidogrel for more than 5 days before surgery might marginally increase postoperative MACCE, while continuing it seemed to be safe and did not increase bleeding risk and transfusion rate.

The study has some limitations. First, the retrospective nature of the study potentially included various information and bias. For example, individual surgeons have different considerations on days of withholding or continuing antiplatelet therapy. The modalities of postoperative thromboprophylaxis and periarticular cocktail formula also varied among different surgeons. The second limitation was the higher proportion of THA (26.7% vs. 11.3%) in group 1. TKA patients used to have more wound complications than THA patients. Therefore, a subgroup analysis was done by comparing only TKA patients in both groups regarding wound complications and infection rates to assure similar results (data not shown). The third confounding variable was that there were more patients in group 1 receiving general anesthesia than group 2 (94.6% vs. 83.3%). It was mostly due to the continuing use of clopidogrel in group 1 against the use of spinal anesthesia. However, more than 80% of patients in both groups received the same type of anesthesia, and the exact influence of the type of anesthesia on arthroplasty patients regarding blood loss and complications was still controversial [39,40]. Lastly, the small sample size and event numbers limited the power of the study and increased the chance of type II error. That being said, the rarity of this kind of population in joint arthroplasty still makes our study meaningful and practical for the orthopedic surgeons to take as a reference when managing patients on clopidogrel. A multi-center study may be needed in the future to further investigate the safety of perioperative antiplatelet/anticoagulant therapy.

## 5. Conclusions

The clinical decisions regarding the management of antiplatelet therapy in arthroplasty patients are challenging and require collaborative approaches. Surgeons must balance the risk of thromboembolism against postoperative complications on an individual basis to optimize outcomes. The results of our study suggested that clopidogrel continuation could be safe and advisable for patients at thrombotic risk undergoing primary major joint replacement. However, well-designed and adequately powered studies including a patient risk-stratification algorithm on thrombosis and hemorrhage are required to prove this suggestion in the future.

## Figures and Tables

**Figure 1 jcm-11-01754-f001:**
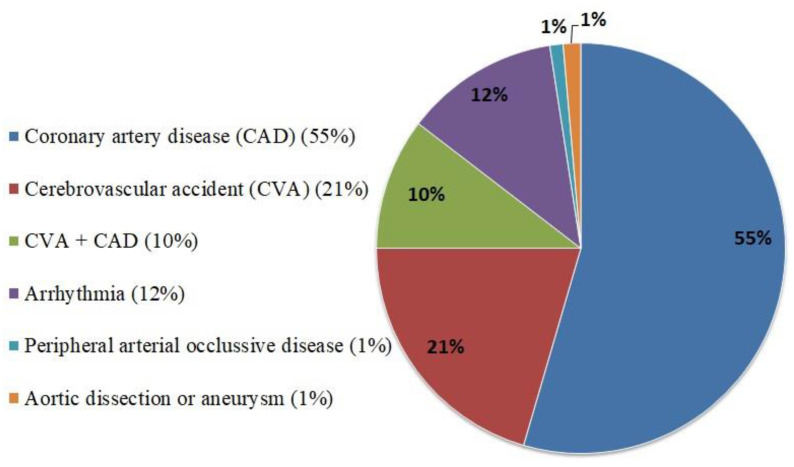
Main indications of clopidogrel in the study patients.

**Table 1 jcm-11-01754-t001:** Demographic data of the study patients.

	Total Patients (*N* = 296)		*p*-Value
	Group 1 (*N* = 56)	Group 2 (*N* = 240)	
**Gender (M/F)**	35/21	103/137	0.008
ASA score ≥ 3	48 (85.7%)	203 (84.6%)	0.83
Mean age (range)	71.5 (40 to 85)	72.6 (50 to 90)	0.73
BMI ≥ 25	39 (69.6%)	185 (77.1%)	0.24
Thrombocytopenia (<150 k)	9 (16.1%)	26 (10.8%)	0.27
PT (INR) (range)	0.99 (0.89 to 1.13)	0.99 (0.87 to 1.19)	0.19
aPTT (range)	27.4 (23.8 to 33.4)	27.6 (21.4 to 36.1)	0.41
Anesthesia (spinal/general)	3/53	40/200	0.03
Charlson score ≥ 3	33 (58.9%)	131 (55.6%)	0.56
CHA2DS2-VASc score (range)	3.52 (1 to 6)	3.55 (1 to 7)	0.96
HAS-BLED score (range)	3.28 (2 to 5)	3.18 (2 to 5)	0.29
Comorbidities			
COPD	3	6	0.26
Cancer	4	11	0.43
Chronic heart failure	2	12	0.65
Chronic renal insufficiency	7	53	0.11
End-stage renal disease	5	6	0.02
Cerebrovascular disease	20	69	0.31
Coronary artery disease	37	150	0.62
Arrhythmia/Af	13	22	0.003
Chronic liver disease	6	17	0.36
Diabetes mellitus	23	109	0.56
Hypertension	56	234	-
Duration of surgery >75th%	15	53	0.67
Post-op VTE prophylaxis			
Rivaroxaban	2	46	
Enoxaparin	6	13	
Pre-op hospital stay (range)	1.4 (1 to 5)	1.2 (1 to 5)	0.67
Post-op hospital stay (range)	4.5 (2 to 10)	5.2 (2 to 27)	0.16
THA/TKA	15/41	27/213	0.002

ASA, American Society of Anesthesiologists; BMI, body mass index; COPD, chronic obstructive pulmonary disease; Af, atrial fibrillation.

**Table 2 jcm-11-01754-t002:** Bleeding risk between the two groups of patients.

	Total Patients (*N* = 296)		*p*-Value
	Group 1 (*N* = 56)	Group 2 (*N* = 240)	
**Pre-op Hb (mean and range)**	12.74 (9.7 to 15.8)	12.52 (7.6 to 16.9)	0.37
POD1 Hb (mean and range)	10.83 (8.3 to 13.8)	10.83 (6.2 to 16.2)	0.95
POD2 Hb (mean and range)	9.81 (7.9 to 13.1)	9.89 (6.0 to 14.9)	0.6
POD 3 or 4 Hb	9.79 (7.1 to 12.5)	9.86 (6.4 to 13.6)	0.67
Transfusion blood (N)	11 (19.6%)	40 (16.7%)	0.28
Mean units of transfusion (U)	2.4 ± 0.8	2.4 ± 1.1	0.85
Estimated blood loss (mL)	1212.3 (685.8 to 2811.8)	1068.9 (495.6 to 3294.3)	0.03

**Table 3 jcm-11-01754-t003:** Complications and antibiotic use between the two groups of patients.

	Total Patients (*N* = 296)			*p*-Value
	Group 1 (*N* = 56)	Group 2 (*N* = 240)	Odds Ratio (95% CI)	
**Patients with wound complications**	12 (21.4%)	36 (15%)	1.55 (0.74~3.21)	0.23
Prolonged discharge	5	17	1.29 (0.45~3.65)	0.58
Hemarthrosis	4	11	1.6 (0.49~5.23)	0.5
Blister formation	3	13	0.99 (0.27~3.59)	0.9
Skin necrosis	2	4	2.19 (0.39~12.24)	0.32
PJI	2 (3.6%)	4 (1.7%)	2.16 (0.39~12.24)	0.32
Duration of anti-prophylaxis (>24 h)	2 (3.6%)	19 (7.9%)	0.43 (0.1~1.91)	0.39
Addition of oral antibiotic after discharge	6 (10.7%)	26 (10.8%)	0.99 (0.39~2.53)	1
MACCE	1 (1.8%)	13 (5.4%)	0.32 (0.04~2.48)	0.48
Stroke/TIA	0	6		
AMI	1	7		
DVT	0	0		
Pulmonary embolism	0 (0%)	1 (0.4%)		
GI bleeding	0 (0%)	8 (3.3%)		
Return to theater	1 (1.8%)	1 (0.4%)	4.35 (0.27~7.1)	0.34
Return to ER (30 days)	7 (12.5%)	34 (14.2%)	0.87 (0.36~2.07)	0.83
90-day mortality	0 (0%)	1 (0.4%)		

PJI, periprosthetic joint infection; MACCE, major adverse cardiocerebral event; TIA, transient ischemic attack; AMI, acute myocardial infarction; DVT, deep vein thrombosis; GI, gastrointestinal; ER, emergency room.

## Data Availability

Data can be obtained from the corresponding author upon reasonable request.

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
