# Peer review of "The Risk of Bleeding and Adverse Events with Clopidogrel in Elective Hip and Knee Arthroplasty Patients"

_jcm, 2022, doi:10.3390/jcm11071754_

Round 1

Reviewer 1 Report

This is a very well done and very topical paper

Difficulties and weaknesses are well defined by the authors and are indeed difficult for all of us 

I believe they have covered all the potential weaknesses of the paper 

most would be corrected by a larger study but difficult to perform

This may be one of the best to date !

This may stimulate a well done muti-center randomized prospective study 

It will be interesting for individual readers plus excellent for journal clugs and reading clubs 

well done!!

Author Response

Thanks for the reviewer's comment and appreciation of our study. We are going to organize a multi-center retrospective study in the coming future to further validate the risk and outcomes regarding clopidogrel use in major joint arthroplasty.   

Reviewer 2 Report

This study investigating the effect of clopidogrel usage during operation retrospectively performed in a large mass arthroplasty surgery hospital. There were no randomization of clopidogrel discontinuation during THA and TKA surgery. In the results, there were no difference in number of transfusion or mean units of transfusion in both groups. However, estimated blood loss in the group of no hold of clopidogrel group was clearly larger.

There is no need to revise the manuscript. But, please describe the number of surgeons and usage of preoperative blood banking and Cell Saver refining of blood shed during operation.

Author Response

Thanks for the reviewer's comment and appreciation of our study. There were six surgeons involved in the study. Two of them didn't hold clopidogrel before TJA, and the other four surgeons hold clopidogrel 5~7 days before TJA. The preparation of blood before surgery was done as routine in our hospital. All the blood transfused was allogeneic from the blood bank of our hospital. We didn't use Cell saver in joint arthroplasty. 

Reviewer 3 Report

This study is interesting dealing with controversial issues which is important to arthroplasty surgeons for everyday practice. We are seeing more and more clopidogreal users who need orthopedic surgical procedures. But there is no firm criteria for perioperative use of this drug weighing risk and benefit. This paper is well written about this issues.

Some comments and questions:

  1. Group 1 is heterogenous mixing continuous user and within 5 days stop user. In terms of pharmacodynamic of clopidogrel, do you think low bleeding related complications than other study in group 1 can be partially due to mixture of this within 5 days short term stop patients? You need explanation.
  2. For inclusion of end stage renal disease, incidence is higher in group 1. Can you explain their effect to adverse outcome?
  3. Conclusion: In last sentence, change to "But well-designed and adequately powered-------are required to prove this suggestion in the future."
  4. In abstract, you need to add sentence saying 'But we need further study to prove this suggestion------'. Because you don't have enough evidence with powerful statistical data to support your suggestion. 

Author Response

Thanks for the reviewer's comment and opinions. Here's the point-by-point responses to the reviewer's questions. 

  1. The design of the study to distinguish the two groups of patients by 5 days of drug cessation was referred to the current guidelines and the literature. The FDA and some research also showed that the platelets returned to normal function by at least 5 days after stopping clopidogrel. In the group 1, there were more than half of the patients continuing clopidogrel without cessation (30 patients continued vs 26 patients held within 5 days). We have done a sub-group analysis within group 1, and the results showed that the complication rates (0.17 vs 0.27, p = 0.35) and blood loss (1289.2 vs 1123.5 ml, p = 0.3) were not significantly different between them.  So the mixture of these patients in group 1 was less likely the reason to result in lower bleeding and complication rates. Of course it will be better and more interesting to further separate these two subgroups to compare with those who held drug for more than 5 days. However, the small patient numbers without adequate statistical power limited our design.
  2. Thanks for the reviewer's question. ESRD has been shown to result in higher complication rates and more blood loss in total joint arthroplasty in some studies. But some other studies showed the opposite. In our cohort, there were 5 ESRD patients in group 1 and 6 ESRD patients in group 2. The mean estimated blood loss for these ESRD patients in each group were 1087.7 ml in gourp 1 and 996.4 ml in group 2, without statistically significant difference. These data were also lower than the group mean blood loss. There were 1 patients in group 1 and 2 patients in group 2 having bleeding related wound complications. So it seemed that the ESRD patients in our cohort, especially in group 1, didn't have higher bleeding related risk and complications. One thing noteworthy was that the transfusion rates were higher, 40% in group 1 and 50 % in group 2. This might result from chronic anemia in these ESRD patients. Of course it was difficult to make any conclusion in our study because of limited case numbers. Further investigation is required to validate these observations. 
  3. Thanks for the reviewer's suggestion. The sentence is revised.
  4. Thanks for the reviewer's suggestion. We added the sentence saying " more studies are required in the future to further prove this suggestion." in the end of the abstract.